# A Catalog of *GNI-A1* Genes That Regulate Floret Fertility in a Diverse Bread Wheat Collection

**DOI:** 10.3390/plants13030330

**Published:** 2024-01-23

**Authors:** Shun Sakuma, Yoko Yamashita, Takako Suzuki, Shuhei Nasuda

**Affiliations:** 1Faculty of Agriculture, Tottori University, Tottori 680-8553, Japan; 2Central Agricultural Experiment Station, Hokkaido Research Organization, Naganuma, Hokkaido 069-1395, Japan; yamashita-yoko@hro.or.jp (Y.Y.); suzuki-takako@hro.or.jp (T.S.); 3Graduate School of Agriculture, Kyoto University, Kyoto 606-8502, Japan; nasuda.shuhei.5z@kyoto-u.ac.jp

**Keywords:** grain yield, floret fertility, wheat, core collection, genetic resources

## Abstract

Modifying inflorescence architecture improves grain number and grain weight in bread wheat (*Triticum aestivum*). Allelic variation in *Grain Number Increase 1* (*GNI-A1*) genes, encoding a homeodomain leucine zipper class I transcription factor, influences grain number and yield. However, allelic information about *GNI-A1* in diverse germplasms remains limited. Here, we investigated *GNI-A1* alleles in a panel of 252 diverse bread wheat accessions (NBRP core collection and HRO breeder’s panel) by target resequencing. Cultivars carrying the reduced-function allele (105Y) were predominant in the NBRP panel, whereas the 105N functional allele was the major type in the HRO panel. Cultivars with the 105Y allele were distributed in Asian landraces but not in European genotypes. Association analysis demonstrated that floret fertility, together with grain size, were improved in cultivars in the NBRP core collection carrying the 105Y allele. These results imply that different alleles of *GNI-A1* have been locally selected, with the 105Y allele selected in East Asia and the 105N allele selected in Europe.

## 1. Introduction

Improving grain yield is one of the most urgent tasks of wheat (*Triticum aestivum* L.) breeding. Grain yield has increased over the past decades due to modifications in plant architecture traits such as plant height [1], inflorescence number [2], and inflorescence shape [3]. A study of a panel of elite winter wheat cultivars released in Western Europe during the past 50 years revealed that the number of grains per spike, grains per m^2^, and total plant biomass showed the highest positive correlations with grain yield [4]. The wheat inflorescence (spike) is composed of alternate spikelets that produce multiple florets [5]. The success in floret development and fertilization determines the final number of grains, thereby affecting grain yield.

*Grain Number Increase I* (*GNI1*) encodes a homeodomain leucine zipper class I transcription factor that regulates floret fertility in wheat [6]. A reduced-function allele of *GNI1* on chromosome 2A (*GNI-A1*) promotes the fertility of apical florets (such as the third and fourth florets) and enhances final grain yield without reducing grain size.

*GNI1* is an ortholog of barley (*Hordeum vulgare* L.) *Six-rowed spike 1* (*Vrs1*), which regulates floret fertility in lateral spikelets [7]. *GNI1*/*Vrs1* evolved through gene duplication specific to the Triticeae tribe [6]. During the domestication of barley, loss-of-function alleles of *Vrs1* (*vrs1.a1*, *a2*, *a3*, and *a4*) were selected by humans at least four times independently [7,8]. In wheat, the reduced-function allele 105Y appeared after the domestication of durum wheat (*Triticum turgidum* ssp. *durum*) and was predominantly selected by breeders [6,9]. A resequencing analysis of *GNI-A1* in 111 genotypes of tetraploid wheat identified eight haplotypes from wild emmer (*T. turgidum* ssp. *dicoccoides*) and two haplotypes each from domesticated emmer and durum wheat [9]. The N105Y mutation was specifically found in durum wheat. Haplotype analysis, using a panel of 210 European winter accessions, revealed that the functional allele (105N) is dominant in this population compared to the reduced-function allele (105Y). Cultivars carrying the 105Y allele showed more grains per spikelet than those without that allele. However, the allelic variation of *GNI-A1* in a diverse wheat panel has not been catalogued to date.

The National Bioresources Project-Wheat (NBRP-Wheat), a gene bank organized by wheat researchers, developed a core collection of 188 genotypes representing a comprehensive collection of worldwide hexaploid wheat [10]. The collection includes classical taxonomic species (*T. aestivum*, *T. spelta*, *T. compactum*, *T. sphaerococcum*, *T. macha*, and *T. vavilovii*), landraces covering the entire geographical distribution of wheat, and modern cultivars grown in Japan. The genetic diversity of the population was analyzed using genome-wide DArTseq markers (13,394 SNPs and 20,523 PAVs). This analysis revealed that the collections are genetically divergent and form at least six groups reflecting the geographical distributions of the genotypes. The Hokkaido Research Organization (HRO) aims to contribute to the improvement of agriculture in the region of Hokkaido, Japan. HRO consists of several breeding stations and a gene bank that maintains germplasm. Here, we investigated the NBRP-wheat core collection together with the HRO wheat collection to explore the allelic variation of *GNI-A1*. We also analyzed the association between this allelic variation and agronomic traits such as floret/grain number per spikelet, spikelet number, tiller number, and days to heading. Our findings shed light on the selection of *GNI-A1* alleles for wheat breeding in both East Asia and Europe.

## 2. Results

### 2.1. Allelic Variation of GNI-A1

We analyzed two panels (NBRP core collection, 188 accessions, and HRO collection, 64 accessions) for allelic variation at the *GNI-A1* locus. We classified the sequence encoding the homeodomain of the encoded homeodomain leucine zipper class I transcription factor, because variation at the 105th amino acid (which may be N, Y, or K) contributes to grain number per spikelet/spike. A resequencing of the 188 NBRP panel classified this population into three groups with different alleles; 57, 112, and 12 accessions were assigned to the 105N, 105Y, and 105K groups, respectively, and 7 accessions could not be genotyped due to a lack of PCR amplification (Appendix A). Accessions carrying the 105N allele comprised three *Triticum* species: *aestivum* (41 accessions), *macha* (4), and *spelta* (12) (Table 1). Accessions carrying the 105Y allele comprised five species: *aestivum* (103), *compactum* (3), *spelta* (1), *sphaerococcum* (2), *vavilovii* (2), and synthetic wheat (1). Accessions carrying the 105K allele were relatively rare and were from just two species: *aestivum* (11) and *compactum* (1). No PCR products were obtained from seven accessions from three species: *T. aestivum* (4), *T. compactum* (2), and *T. sphaerococcum* (1). Among the 155 *T. aestivum* accessions genotyped, cultivars carrying 105Y are predominantly distributed in Afghanistan, Bhutan, China, Ethiopia, Iran, Japan, Nepal, and Pakistan, whereas those carrying 105N are distributed in Georgia, the UK, and the USA (Table 2). Although cultivars carrying 105K are relatively rare, they are found in Greece, Iran, Iraq, Japan, Spain, and Turkey.

Genotyping of *GNI-A1* in the HRO panel revealed that 39 accessions contained the 105N allele, 13 contained the 105Y allele, and 12 contained the 105K allele (Appendix A). In this panel, accessions with the 105N allele were predominant (60.9%), and nearly equal numbers of accessions carried the 105Y and 105K alleles (20.3% and 18.8%, respectively). Cultivars carrying the 105N allele are mainly distributed in Europe and the USA (Table 3). Notably, cultivars with the 105K allele are predominant in Japan (mainly in Hokkaido). The HRO panel includes modern cultivars and their genotype information, which could be used to select breeding material in the future.

### 2.2. Association of Agricultural Traits with GNI-A1 Variants

We evaluated agronomically important traits such as the number of florets/grains per spikelet and the number of spikelets per spike in the NBRP panel. We focused on accessions carrying the 105N (57 accessions) and 105Y alleles (112); cultivars with the 105K allele (12) were excluded here due to the smaller size of this group. Notably, cultivars carrying the 105Y allele produced more florets per spikelet at the apical, middle, and basal positions of the spike (4.69, 5.67, and 5.61, respectively) compared to cultivars carrying the 105N allele (4.25, 5.29, and 5.19, respectively) (Figure 1A). Also, cultivars carrying the 105Y allele produced more grains per spikelet at apical, middle, and basal positions of the spike (2.62, 3.50, and 3.38, respectively) than those carrying the 105N allele (2.11, 2.97, and 2.87, respectively) (Figure 1B). These results strongly suggest that cultivars with the 105Y allele were preferentially selected due to higher floret fertility.

We also investigated other agricultural traits to understand their relationships with the *GNI-A1* alleles (Figure 2). Cultivars carrying 105Y showed earlier days to heading (182.9 on average) and fewer tillers per plant (15.9) compared to those carrying 105N (195.2 and 17.6, respectively). Lower spike length and spikelet number per spike were also observed in cultivars carrying the 105Y allele (106.0 and 17.5) vs. the 105N allele (120.1 and 19.4). Hundred-grain weight was nearly 10% higher (3.64) in the 105Y cultivars than in the 105N cultivars (3.28). No significant difference in culm length was observed. In addition, we observed significant correlations between days to heading and other traits (Figure 3). Notably, days to heading accounted for approximately 54% of the observed variation in spikelet number per spike (*R*^2^ = 0.54), 32% for spike length and 35% for culm length.

## 3. Discussion

In this study, we genotyped the *GNI-A1* locus in 181 wheat accessions in the NBRP core collection and 64 wheat accessions in the HRO breeder’s collection. Information about the genotypes at the *GNI-A1* locus will facilitate more efficient decision-making in the selection of breeding materials. Notably, cultivars carrying the 105Y allele were dominant in the NBRP collection. This is the opposite case compared to the HRO collection and the European winter wheat collection (mainly from Germany and France), as previously reported [6]. As in the previous study, we observed that cultivars carrying the 105Y allele showed better floret fertility. Among the six classical species, genotypes harboring the 105Y allele were also predominant, except in *spelta*. A previous study also found that most *durum* wheat possesses the 105Y allele. In the NBRP hexaploid panel, the 105Y allele was predominant in cultivars from Asian countries such as Japan, China, and Iran, pointing to the local adaptation of the gene. These results indicate that the 105Y allele was preferred in the bread wheat population after the divergence of *durum* and the transmission of hexaploid wheat from the Middle East to East Asia.

We investigated the association between the *GNI-A1* allele and agronomic traits in the NBRP panel, although the phenotypic data used in this study were obtained from a single experiment. Considering the potentially variable effects of *GNI1* on agronomic traits under different environments with the currently unknown genotype-by-environment interactions, evaluations under three or more environments would be necessary for future work. Notably, the 105Y carriers showed better performance than the 105N carriers in terms of spike length, number of spikelets per spike, and hundred-grain weight. By contrast, days to heading and number of tillers per plant were lower in the 105Y cultivars. A previous study showed that knocking down *GNI1* expression by RNA interference had no significant effects on the transgenic plants in terms of plant height, spike number, spike length, spikelet number, or grain size [6]. These findings indicate that the function of the *GNI1* gene is likely organ-specific and that the phenotypic effects observed in this current study could be due to effects from other genes linked to *GNI1*. The two alleles appear to correspond to two spike shapes: the floret-fertility-oriented type (105Y) and the spikelet-number-oriented type (105N). Notably, these types of alleles seem to follow a geographical distribution: Asian cultivars preferentially exhibit the former type, and European cultivars the latter type. Considering that the climate in the Hokkaido region is relatively cool, like that in the winter-wheat-growing regions of Germany and France, days to heading can be longer in these areas than in other parts of Japan. We detected a positive correlation between days to heading and number of spikelets per spike (Figure 3). An increase in the number of spikelets can ensure a decrease in the number of fertile florets within a spikelet, and therefore the 105N allele could have been preferred.

The final number of wheat spikelets is determined at the terminal spikelet stage when the spike is approximately 3–5 mm long [11]. Recent studies have revealed that the number of spikelets is controlled by *WHEAT ORTHOLOG OF APO1* (*WAPO-A1*/*TaAPO-A1*), which encodes an F-box protein and is located on the long arm of chromosome 7A [12,13]. The expression level of this gene is positively correlated with the number of spikelets per spike [14]. Mizuno et al. [15] demonstrated that combining favorable alleles of both *GNI-A1* and *WAPO-A1* contributes to grain number per spike. The authors also determined that days to heading affects both grain number and grain size. Furthermore, supernumerary/paired spikelet (SS) formation is highly associated with flowering genes such as *PHOTOPERIOD RESPONSE LOCUS1* (*PPD-1*) and *FLOWERING LOCUS T 1* (*FT1*) [16]. An increased dosage of wheat *TEOSINTE BRANCHED1* (*TB1*), whose homolog in maize (*Zea mays*) is a suppressor of tillering, promotes the SS phenotype by interacting with FT1 and suppressing tiller outgrowth [17]. These findings strongly suggest that the process determining the final grain number in wheat is rather complicated and difficult to control. However, accumulating knowledge of the genetic basis of the grain number opens new avenues that will enable us to design crops based on several repertoires of genes in the future. In conclusion, our findings on allelic variations of *GNI-A1* in diverse wheat germplasms, together with the association of these variations with agricultural traits, provide novel insights into enhancing the grain number in wheat.

## 4. Materials and Methods

### 4.1. Plant Materials

A total of 252 wheat genotypes from two panels were used in this study. The first panel (NBRP panel) is composed of 188 diverse bread wheat accessions developed by the NBRP-Wheat, which was used for resequencing and phenotyping (Appendix A). The NBRP germplasms are available upon request at https://shigen.nig.ac.jp/wheat/komugi/ (accessed on 16 January 2024). The second panel is a set of 64 accessions selected by the Hokkaido Research Organization (HRO panel), which was used for resequencing (Appendix A). The HRO genetic resources can be found at http://www.agri.hro.or.jp/grdb/index.php (accessed on 16 January 2024).

### 4.2. Resequencing

Genomic DNA was extracted from young leaves using the Doyle and Doyle protocol [18]. The *GNI-A1* (1768 bp) fragment was amplified using the primers 5′-AGTCTCCAAAATTAAGTGGCAT-3′ and 5′-TGCCATTAATACACACTCTCCA-3′. PCR amplification was carried out in 10 μL reactions containing 0.25 U PrimeSTAR GXL DNA polymerase (Takara, Tokyo, Japan), 1× PrimeSTAR GXL buffer, 0.3 μM of each primer, 200 μM dNTP, and 20 ng genomic DNA. Each PCR included a denaturation step (98 °C/5 min) followed by 30 cycles of 98 °C/10 s, 60 °C/15 s, 68 °C/60 s. The PCR products were purified using the NucleoFast 96 PCR Clean-up kit (Macherey-Nagel, Düren, Germany) and subjected to cycle sequencing using a Big Dye Terminator Kit (Applied Biosystems, Foster, CA, USA) with the sequencing primer 5′-GGATGGTAACGGCTGGGAGA-3′. Sequencing reactions were purified using the Agencourt CleanSEQ system (Beckman, Beverly, MA, USA) and analyzed with an ABI prism 3500 genetic analyzer (Applied Biosystems). Sequence data were aligned using Sequencher DNA Sequencing Software version 5.4.5 (Hitachi-Soft, Yokohama, Japan).

### 4.3. Phenotyping

Five plants per accession of the NBRP-wheat core collection were grown in the experimental field in the Arid Land Research Center, Tottori University, Japan. After harvesting, three representative spikes per line were used to measure the number of florets/grains per spikelet along the spike position (apical, middle, and basal). Data on other agronomic traits, such as days to heading, culm length, tiller number, spike length, spikelet number per spike, and hundred-grain weight, were obtained from archived data [10].

## Figures and Tables

**Figure 1 plants-13-00330-f001:**
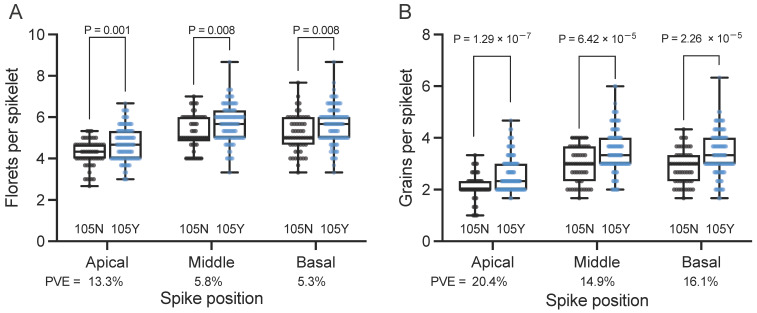
Phenotypic effects of the *GNI-A1* alleles on spikelet morphology in the NBRP collection. (**A**) Number of florets per spikelet after harvesting. (**B**) Number of grains per spikelet. The number of florets/grains per spikelet was determined from the apical, middle, and basal parts of the spike. Black indicates cultivars carrying the 105N allele (*n* = 57; *Triticum aestivum*: 41 accessions, *macha*: 4, and *spelta*: 12) and blue indicates cultivars carrying the 105Y allele (*n* = 112; *aestivum*: 104, *compactum*: 3, *spelta*: 1, *sphaerococcum*: 2, *vavilovii*: 2, and synthetic wheat: 1). *P* values were determined using the Student’s *t*-test. PVE: phenotypic variance explained by *GNI-A1* alleles.

**Figure 2 plants-13-00330-f002:**
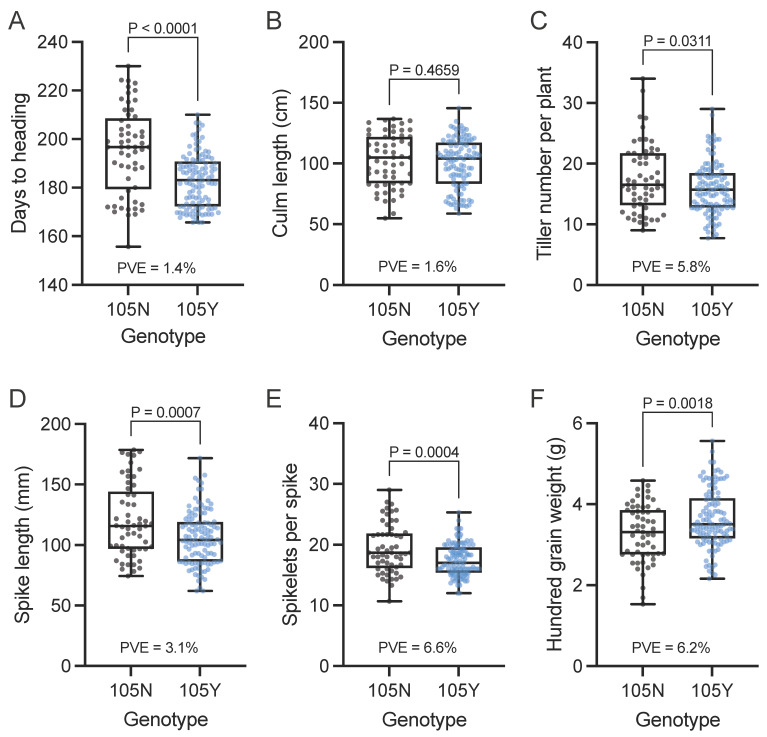
Agronomic traits of members of the NBRP core collection with the 105N or 105Y allele. (**A**) Days to heading, (**B**) Culm length, (**C**) Number of tillers per plant, (**D**) Spike length, (**E**) Number of spikelets per spike, (**F**) Hundred-grain weight. Cultivars carrying the 105N allele include 57 accessions (*Triticum aestivum*: 41 accessions, *macha*: 4, and *spelta*: 12) and cultivars carrying the 105Y allele include 112 accessions (*aestivum*: 104, *compactum*: 3, *spelta*: 1, *sphaerococcum*: 2, *vavilovii*: 2, and synthetic wheat: 1), respectively. *P* values were determined using the Student’s *t*-test.

**Figure 3 plants-13-00330-f003:**
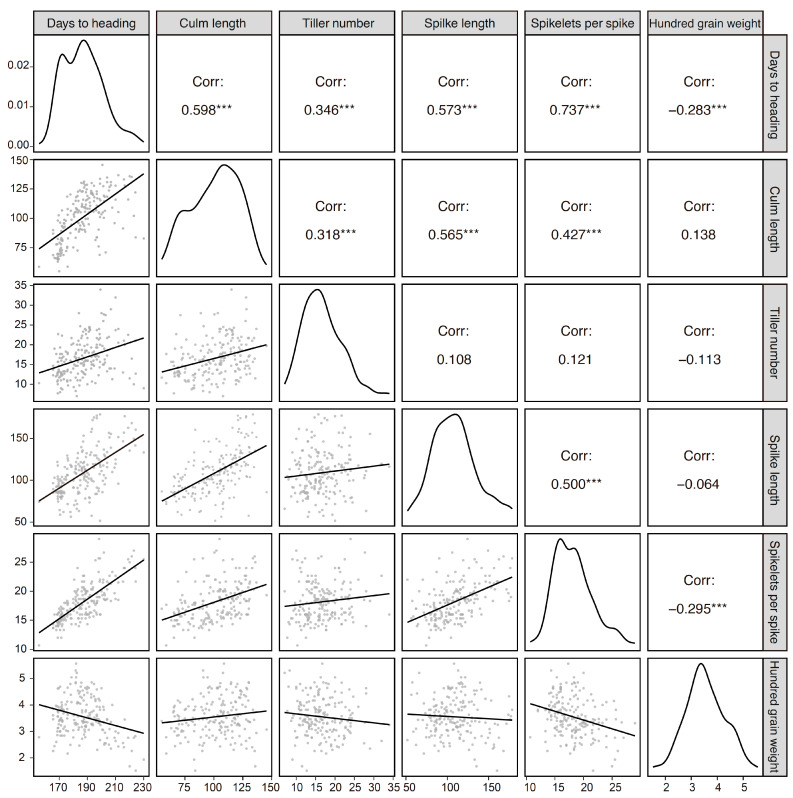
Correlations between agricultural traits. The phenotypic data were obtained from the NBRP panel (*n* = 188; *Triticum aestivum*: 159 accessions, *compactum*: 6, *macha*: 4, *spelta*: 13, *sphaerococcum*: 2, *vavilovii*: 2, and synthetic wheat: 1). Asterisks indicate significant correlations at *p* < 0.001 (***).

**Table 1 plants-13-00330-t001:** Allelic variation of *GNI-A1* in hexaploid wheat in the NBRP core collection.

Genus	Species	105N	105Y	105K	Unknown
*Triticum*	*aestivum*	41	103	11	4
	*compactum*	0	3	1	2
	*macha*	4	0	0	0
	*spelta*	12	1	0	0
	*sphaerococcum*	0	2	0	1
	*vavilovii*	0	2	0	0
	Synthetic	0	1	0	0
	Total	57	112	12	7

**Table 2 plants-13-00330-t002:** Geographic distribution of the *GNI-A1* alleles in *Triticum aestivum* in the NBRP core collection.

Species	Country	105N	105Y	105K	Unknown
*Triticum aestivum*	Afghanistan	1	12	0	1
	Armenia	1	1	0	0
	Australia	0	2	0	0
	Azerbaijan	1	0	0	0
	Bhutan	0	4	0	0
	Canada	0	1	0	0
	China	3	12	0	0
	Egypt	1	0	0	0
	Ethiopia	0	7	0	1
	Georgia	3	0	0	0
	Greece	1	1	2	1
	India	0	1	0	0
	Iran	2	14	1	0
	Iraq	0	1	1	0
	Italy	1	0	0	0
	Japan	5	24	5	0
	Jordan	1	0	0	0
	Lebanon	1	0	0	0
	Mexico	1	1	0	0
	Nepal	1	5	0	0
	Pakistan	0	5	0	0
	Romania	2	3	0	0
	Spain	2	2	1	0
	Syria	1	0	0	0
	Tanzania	0	1	0	0
	Turkey	4	4	1	0
	UK	6	0	0	0
	USA	3	1	0	1
	Uzbekistan	0	1	0	0
	Total	41	103	11	4

**Table 3 plants-13-00330-t003:** Geographic distribution of the *GNI-A1* alleles in *Triticum aestivum* in the HRO panel.

Species	Country	105N	105Y	105K	Unknown
*Triticum aestivum*	Australia	3	0	0	0
	Canada	3	0	0	0
	Germany	4	0	0	0
	Japan	4	1	7	0
	Mexico	1	0	0	0
	Russia	1	0	0	0
	Switzerland	1	0	0	0
	Turkey	0	0	1	0
	UK	6	6	0	0
	USA	15	6	4	0
	Yugoslavia	1	0	0	0
	Total	39	13	12	0

## Data Availability

The original data presented in the study are included in the article/supplementary materials, further inquiries can be directed to the corresponding author.

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
