# Peer review of "A Catalog of GNI-A1 Genes That Regulate Floret Fertility in a Diverse Bread Wheat Collection"

_plants, 2024, doi:10.3390/plants13030330_

Round 1

Reviewer 1 Report

Comments and Suggestions for Authors

This manuscript is well structured and described. I have comments regarding to this manuscript, which are rather minor.

Figure 1, 2 and 3, please clarify the taxonomic species used for phenotypic analysis. As mentioned, several taxonomic species was used for the GNI-A1 diversity analysis. I am wondering if it’s reasonable to include all samples from different sub-species which are significantly diversified.

Figure 1B, p value. Please use the exact value, in order to be accordant with that of Figure 1A.

It might be better to move Tables 1-5, ahead of Figures.

Reviewer 2 Report

Comments and Suggestions for Authors

This manuscript describes allelic variation of wheat GNI-A1 gene in two hexaploid wheat panels composed of diverse accessions. It also reports the association between the GNI-A1 alleles and several agronomic traits including floret/grain number per spikelet. The manuscript is well-written overall. However, I have several major/minor suggestions to improve the interpretation of results and presentation of data.

Major points:

- Line 164-166: I am concerned of the authors’ interpretation of the association between GNI1 and agronomic traits (i.e., days to heading, tiller number, spike length, spikelet number, grain weight) other than floret/grain number. The authors suggest that this association could be from a pleiotropic control by GNI1. This is not consistent with the author’s previous findings in reference [6]. It is also highly plausible that other genes linked to GNI1 may affect different agronomic traits.

- In addition, the phenotype data used in this study is from a single field experiment for florets/grains per spike, and from a single glasshouse experiment with plants grown in pots in reference [10] for other traits. This stands out as the major limitation of this study. Considering the potentially variable effects of GNI1 on agronomic traits under different environments with the currently unknown genotype-by-environment interactions, evaluations under three or more environments would be necessary. If this is not feasible, I think it is crucial to at least acknowledge and discuss this limitation thoroughly in the discussion section.

Minor points:

- Figure 1: There is no indication for 105N and 105Y alleles in the figure, while the legend mentions different colors (black and blue) for the indication.

- For each trait in figure 1 and 2, could you add how much phenotypic variance is explained by GNI-A1?

- Figure 3: What about presenting a correlation matrix for all possible pairs of the agronomic traits, instead of selectively presenting the correlation between days to heading and other traits?

- I think Table 1 and Table 4 are too lengthy to put in the main manuscript. I suggest putting them as supplementary tables.

- Is Table 3 complete and correct? I see the total number of accessions is only 159 (41+103+11+4) in Table 3, while it is 188 in Table 1 and Table 2.

Comments on the Quality of English Language

na

Reviewer 3 Report

Comments and Suggestions for Authors

In the present study, authors examined allelic variation of GNI-A1 alleles of Triticum aestivum in a panel of 252 diverse bread wheat accessions. The study is well designed and methods are suitable. The results are confirmatory of related studies in other collections and of interest for wheat breeders.

The manuscript is well-writen.

All tables need clarification in the legends

Table 1,4: define column GNI-1A allele as in line 71. explain  dash (-) symbol

Table 2,3,5. align columns

Round 2

Reviewer 2 Report

Comments and Suggestions for Authors

All comments and suggestions have been addressed properly.

Comments on the Quality of English Language

na